# Deregulated Transcription and Proteostasis in Adult *mapt* Knockout Mouse

**DOI:** 10.3390/ijms24076559

**Published:** 2023-03-31

**Authors:** Pol Andrés-Benito, África Flores, Sara Busquet-Areny, Margarita Carmona, Karina Ausín, Paz Cartas-Cejudo, Mercedes Lachén-Montes, José Antonio Del Rio, Joaquín Fernández-Irigoyen, Enrique Santamaría, Isidro Ferrer

**Affiliations:** 1Neurologic Diseases and Neurogenetics Group, Bellvitge Institute for Biomedical Research (IDIBELL), 08907 L’Hospitalet de Llobregat, Barcelona, Spain; 2CIBERNED (Network Centre of Biomedical Research of Neurodegenerative Diseases), Institute of Health Carlos III, 08907 L’Hospitalet de Llobregat, Barcelona, Spain; 3Neuropharmacology & Pain Group, Pharmacology Unit, Department of Pathology and Experimental Therapeutics, School of Medicine and Health Sciences, Institute of Neurosciences, University of Barcelona, 08907 L’Hospitalet de Llobregat, Barcelona, Spain; 4Neuropathology Group, Bellvitge Institute for Biomedical Research (IDIBELL), 08907 L’Hospitalet de Llobregat, Barcelona, Spain; 5Clinical Neuroproteomics Unit, Proteomics Platform, Proteored-ISCIII, Navarrabiomed, Complejo Hospitalario de Navarra (CHN), Universidad Pública de Navarra (UPNA), diSNA, 31008 Pamplona, Navarra, Spain; 6Molecular and Cellular Neurobiotechnology Group, Institute of Bioengineering of Catalonia (IBEC), Barcelona Institute for Science and Technology, Science Park Barcelona (PCB), 08028 Barcelona, Barcelona, Spain; 7Department of Pathology and Experimental Therapeutics, University of Barcelona, 08907 L’Hospitalet de Llobregat, Barcelona, Spain; 8Emeritus Researcher, Bellvitge Biomedical Research Institute (IDIBELL), Emeritus Professor, University of Barcelona, 08907 L’Hospitalet de Llobregat, Barcelona, Spain

**Keywords:** tau-KO, transcriptomics, phosphoproteomics, cytoskeleton, synapse

## Abstract

Transcriptomics and phosphoproteomics were carried out in the cerebral cortex of *B6.Cg-Mapttm1(EGFP)Klt* (tau knockout: tau-KO) and wild-type (WT) 12 month-old mice to learn about the effects of tau ablation. Compared with WT mice, tau-KO mice displayed reduced anxiety-like behavior and lower fear expression induced by aversive conditioning, whereas recognition memory remained unaltered. Cortical transcriptomic analysis revealed 69 downregulated and 105 upregulated genes in tau-KO mice, corresponding to synaptic structures, neuron cytoskeleton and transport, and extracellular matrix components. RT-qPCR validated increased mRNA levels of *col6a4*, *gabrq*, *gad1*, *grm5*, *grip2*, *map2*, *rab8a*, *tubb3*, *wnt16*, and an absence of *map1a* in tau-KO mice compared with WT mice. A few proteins were assessed with Western blotting to compare mRNA expression with corresponding protein levels. *Map1a* mRNA and protein levels decreased. However, β-tubulin III and GAD1 protein levels were reduced in tau-KO mice. Cortical phosphoproteomics revealed 121 hypophosphorylated and 98 hyperphosphorylated proteins in tau-KO mice. Deregulated phosphoproteins were categorized into cytoskeletal (*n* = 45) and membrane proteins, including proteins of the synapses and vesicles, myelin proteins, and proteins linked to membrane transport and ion channels (*n* = 84), proteins related to DNA and RNA metabolism (*n* = 36), proteins connected to the ubiquitin-proteasome system (UPS) (*n* = 7), proteins with kinase or phosphatase activity (*n* = 21), and 22 other proteins related to variegated pathways such as metabolic pathways, growth factors, or mitochondrial function or structure. The present observations reveal a complex altered brain transcriptome and phosphoproteome in tau-KO mice with only mild behavioral alterations.

## 1. Introduction

The microtubule-associated proteins tau are encoded by the tau gene *MAPT* or *mapt* and are localized in the cytoplasm and nucleus of the cells. These proteins are mainly abundant in the neurons of the central nervous system (CNS) and are less expressed in brain-resident immune cells: astrocytes and oligodendrocytes [1]. In the cytoplasm, tau primarily maintains the stability of microtubules in axons and participates in cellular polarity and the anterograde and retrograde axonal transport of organelles and vesicles. Tau interacts with tubulin, actin, and many proteins and lipids in the cytoplasm, cell membranes, and synapses. In the nucleus, tau interplays with DNA and proteins in DNA protection. 

The adult human brain expresses at least six isoforms of tau proteins derived from the alternative splicing of its messenger RNA. The tau isoforms range in size from 352 to 441 amino acid residues and differ in the presence of exons 2 and 3, which encode N-terminal sequences, and exon 10, which encodes a microtubule-binding repeat domain. Exclusion of exon 10 results in isoforms with three microtubule binding domain repeats (3R), while inclusion results in isoforms with four repeats (4R) [2]. The presence of each tau isoform differs in the human and rodent central nervous systems. 3R tau predominates during brain development in both humans and rodents. 4R tau increases during early life and reaches equal levels to 3R in adulthood in humans at a 1:1 ratio [3,4], whereas 4R tau isoforms predominate in the brains of adult mice [5,6,7,8,9,10,11,12,13,14,15,16,17,18]. 

In humans, tau is subject to many post-translational modifications, including hyperphosphorylation, truncation, nitration, glycation, glycosylation, ubiquitination, and polyaminations, that may affect its conformational disposition and tau self-aggregation capacity [19,20,21,22]. Generally, an appropriate amount of phosphorylation is necessary for the realization of physiological functions of tau, whereas the hyperphosphorylated state reduces its biological activity. As a natively unfolded protein, tau presents little tendency for aggregation. Evidence has revealed that the tau molecule might be prone to conformation change to make a paperclip-like structure in the presence of the N-terminal, C-terminal, and microtubule-binding domain (MBD) [23]. The paperclip-like shape might prevent tau aggregation. Once this structure is broken, for example, through hyperphosphorylation, these inhibitory domains of the tau molecule will be neutralized, and then tau protein self-assembles into tangles of paired helicoidal filaments (PHFs) [24,25]. 

Aggregation of abnormally hyperphosphorylated microtubule-binding protein tau in certain populations of neurons and glial cells is a pathological hallmark of several neurodegenerative disorders, such as Alzheimer’s disease (AD) and tauopathies. In these pathological contexts, the normal role of tau protein is ineffective to keep the cytoskeleton well organized in the axonal process due to the loss of its capacity to bind to microtubules. This abnormal behavior is promoted by other conformational changes and misfoldings in the normal structure of tau that lead to its aberrant aggregation into fibrillary structures inside the neurons and glial cells [26,27,28]. Tau aggregation decreases the levels of soluble functional tau, sequesters other cell components, or hinders axonal transport, finally resulting in neurodegeneration. Tauopathies represent a heterogeneous group of around 20 neurodegenerative diseases characterized by abnormal deposition of MAPT in neurons and glial cells [29,30]. Histopathologically, the tauopathies are characterized by the presence of intracellular, insoluble inclusions of abnormally modified tau protein in neurofibrillary or gliofibrillary tangles. 

Several transgenic animals have been generated to learn about *mapt* mutations leading to tau aggregation and toxic gain of function [31,32]. Some other models have explored the effects of tau ablation and loss of function [33,34,35,36,37]. However, the latter have been less studied, and there is no general agreement about the clinical/behavioral manifestations of the tau-KO genotype in mice. Initial reports described a mild or absent phenotype in tau-KO mice; subsequent studies disclosed a discrete and inconclusive phenotype when assessing motor performance, fear conditioning, anxiety, and learning [38]. Discrepancies may be due to specific characteristics of the different tau-KO strains, environmental factors linked to the care and diets of mice, variability in the age of the mice at examination, and differences in the interpretation of relatively subtle changes in these behavioral responses. Yet, commonly accepted characteristics of tau-KO mice are the reduced response to anxiety and stress [39,40,41,42] and the protection against drug-induced seizures and ischemic damage upon middle cerebral artery occlusion with reperfusion [37,43,44,45,46]. Protection in tau-KO mice is associated with reduced neuronal excitability [47,48], albeit long-term hippocampal potentiation and long-term depression are variable depending on the tau-KO strain [49,50,51]. The role of tau is also manifested in the modulation of adult neurogenesis in the hippocampus, olfactory bulb, and subventricular region under prolonged stress [52,53]. Tau deletion also prevents stress-induced dendritic atrophy [54].

In addition to these data, a few transcriptome and phosphoproteome studies are available in tau-KO mice, and most of them are restricted to selected genes and proteins [39,42,55,56]. The present work was designed to obtain combined information on altered whole cortex brain mRNA and protein expression in aged tau-KO mice using robust transcriptomics and phosphoproteomics technologies and results validation by current RT-qPCR and Western-blot techniques, in addition to mouse model behavioral evaluation. 

## 2. Results

### 2.1. Molecular Characterization of Tau Protein and Behavioral Status of Tau-KO Mice

Previous results from our group showed that the telencephalon of WT was not stained with anti-3R tau antibodies except for polymorphic cells of the inner region of the dentate gyrus and a few neurons in the entorhinal cortex. The neuropil of the telencephalon showed weak and diffuse 4R tau immunoreactivity in WT animals, whereas anti-PHF1 antibodies revealed moderate immunoreactivity in the neuropil but not in the cell bodies in WT animals. In contrast, tau-KO mice showed negative 3R tau, 4R tau, and PHF1 immunoreactivity in the telencephalon [57]. In the present work, Western blots of brain homogenates of WT mice confirmed previous results and showed immunoreactive bands at the appropriate molecular weight for antibodies tau 5, 4R tau, and PHF1, whereas 3R tau was negative, probably due to the dilution of the small amounts of isolated positive cells in the total cortex homogenates of WT mice. In contrast, Western blots to tau 5, 4R tau, and PHF1 (and 3R tau) of brain homogenates from tau-KO mice were negative (Figure 1A). Densitometric studies further validated the significant differences between WT and tau-KO mice (*p* < 0.001) (Figure 1A). 

Tau-KO mice did not show any obvious physical or behavioral alterations. Moreover, no differences between WT and tau-KO mice were observed in the two-object recognition test (Figure 1B). The elevated plus maze revealed a reduced anxiety-like behavior in tau-KO mice, which spent more time in the open arms compared with WT (*p* < 0.05) (Figure 1C). In both tests, WT and tau-KO animals displayed similar exploration times, suggesting that spontaneous activity is not affected by tau ablation. Regarding the fear conditioning test, both genotypes displayed similar freezing levels during aversive contextual conditioning (Figure 1D), but tau-KO mice showed lower freezing levels than WT mice during re-exposure to the aversively conditioned context (*p* < 0.05) (Figure 1E).

### 2.2. Transcriptomic Alterations in tau-KO Animals Using RNA-seq

Transcriptomic analysis using RNA-seq revealed differentially expressed genes in tau-KO mice compared with WT. Age and sex were not relevant for the analysis. After filtering, 12,410 genes were not significantly altered; 69 were downregulated and 105 were upregulated. Deregulated transcripts are graphically illustrated in the heat map representation at a probability of an adjusted *p*-value < 0.05 (Figure 2A). Enrichment analysis against the Gene Ontology database (*p* < 0.05) identified the main altered genes corresponding to (i) synaptic structures, (ii) neuron cytoskeleton and transport, and (iii) extracellular matrix components (Figure 2B). Deregulated genes are listed in Appendix A.

### 2.3. Gene Expression Validation

Based on the RNA-seq results, genes of different components of the altered pathways revealed by transcriptomics were evaluated using RT-qPCR, including *col6a4*, *gabrb2*, *gabrq*, *grip2*, *mapt*, *nlgn3*, *rab37*, *utrn*, and *wnt16*. In addition, mRNA expression was assessed for other genes linked with neurotransmitters and the cytoskeleton: *gad1*, *grm5*, *map1a*, *map2*, *rab8a*, and *tubb3.* Tau-KO mice showed increased mRNA levels of *col6a4* (*p* = 0.000), *gabrq* (*p* = 0.000), *gad1* (*p* = 0.03), *grm5* (*p* = 0.028), *grip2* (*p* = 0.04), *map2* (*p* = 0.039), *rab8a* (*p* = 0.000), *tubb3* (*p* = 0.01), and *wnt16* (*n* = 0.05) and reduced map1a (*p* = 0.048) compared with WT mice (Figure 3). As expected, *mapt* mRNA expression was 0 in tau-KO mice (Figure 3). Finally, *gabrb2* (*p* = 0.74), *nlgn3* (*p* = 0.09), *rab37* (*p* = 0.14), and *utrn* (*p* = 0.14) expression was similar in tau-KO and WT mice (Figure 3).

### 2.4. Phosphoproteome in Tau-KO Mice

Heatmaps of the cerebral cortex’ phosphoproteome showed marked differences between tau-KO and WT mice. A quantity of 219 deregulated phosphoproteins, 121 hypophosphorylated, and 98 hyperphosphorylated were revealed by phosphoproteomics (30% fold-change and *p* < 0.05) (Figure 4A; Appendix A). The altered phosphoproteome in tau-KO mice mainly affected proteins putatively located in synapses (pre-synapses and post-synapses), followed by axons, dendrites, and neuronal cell bodies (Figure 4B).

Deregulated phosphoproteins were manually categorized into several groups according to their putative localization and function, also assessing their phosphorylation direction (hypo- or hyperphosphorylation). Cytoskeletal (*n* = 45) and membrane proteins (*n* = 84), including proteins of the synapses and vesicles, myelin proteins, and proteins linked to membrane transport and ion channels, accounted for 129 deregulated phosphoproteins. In addition, proteins related to DNA and RNA metabolism (*n* = 36); proteins connected to the ubiquitin-proteasome system (UPS) (*n* = 7); proteins with kinase or phosphatase activity (*n* = 21); and 22 other proteins related to variegated pathways such as metabolic pathways, growth factors, or mitochondrial function or structure completed the list of deregulated phosphoproteins in tau-KO mice (Appendix A).

### 2.5. Protein Validations Using Immunoblotting

Western blotting was used to assess the protein levels of microtubule associate protein 1a (MAP1A), β-tubulin III, and glutamate decarboxylase 1 (GAD1). A significant reduction of MAP1A (*p* = 0.011), β-tubulin III (*p* = 0.000), and GAD1 (*p* = 0.008) was found in tau-KO compared with WT mice. Regarding phosphorylated proteins, phosphorylated neurofilament heavy chain (pNFL-H) was increased (*p* = 0.012) and phosphorylated neurofilament light chain (pNFL-L) decreased (*p* = 0.011) in tau-KO mice (Figure 5).

## 3. Discussion

As tau protein plays a crucial role in human brain development and function, it is expected that the tau-KO mouse displays a severe phenotype. However, homozygous tau-KO mice develop normally and do not display any overt histological abnormalities. The lack of tau protein in the mouse model has resulted in subtle changes at different levels that need to be considered. No physical variation has been observed; however, subtle changes have been reported at the behavioral level. The present study further confirms particular changes in the phenotype of tau-KO mice when compared to WT animals, including reduced anxiety-like behavior as observed in the elevated plus maze. Tau-KO mice also show lower fear expression levels during re-exposure to a threat-conditioned context, which might suggest impaired consolidation of aversive memories. Since this task relies on both hippocampal and amygdalar function, this result might point to altered memory processing in these areas. However, unaltered tau-KO mice’s performance in the novel object recognition test, a hippocampus-dependent task, discards deficiencies in the acquisition, consolidation, or retrieval of non-emotional memories. Thus, reduced fear expression after contextual fear conditioning is more likely linked to an abnormally diminished expression of fear and/or anxiety-associated responses than to cognitive disruption. These results agree with previous observations of anxious behaviors in tau-KO mice, which described how tau-KO mice did not develop any stress-induced depressive behaviors or deficits of hippocampal function compared to C57BL/6 mice [58].

The relative subtlety of the tau-KO phenotype is probably partly due to developmental plasticity and redundancy associated with the different microtubule-associated proteins present in the brain. A reproducible molecular characteristic in different tau-KO transgenic mice and hippocampal neurons from tau-deficient mice is the increased expression of MAP1A, MAP1B, and MAP2 in newborn and middle-aged [33,34,36,59,60]. However, expression of MAP1A, MAP1B, and MAP2 decreases in aged tau-KO mice [59]. Our study shows reduced *map1a* RNA and MAP1 protein levels but increased *map2* mRNA expression in tau-KO mice aged 12 months. Increased MAP2 may compensate for functions linked to other microtubule-associated proteins [60].

The present transcriptomics study identifies 69 downregulated, 105 upregulated, and 12,410 genes not significantly altered in tau-KO mice. Enrichment analysis against the Gene Ontology database categorizes the main altered genes implicated in (i) synapsis, (ii) neuron cytoskeleton, and (iii) extracellular matrix components. Regarding deregulated synaptic components, increased mRNA expression levels of γ-aminobutyric acid type A receptor subunit theta (*gabrq*), glutamate decarboxylase (*gad1*), and metabotropic glutamate receptor 5 (*grm5*) are validated with RT-qPCR in tau-KO mice. Altered cytoskeleton and neuronal components are supported by increased mRNA expression of the microtubule-associated protein 2 gene (*map2*) and RAB8A, a member of the RAS oncogene family (*rab8*) involved in the transport of proteins from the endoplasmic reticulum to the Golgi complex and the plasma membrane, together with reduced *map1a* and zero levels of the microtubule-associated protein tau (*mapt*), as expected. Increased collagen type VI (*col6a4*) mRNA supports the altered expression of extracellular matrix components in tau-KO mice. Increased expression of *gabrb2*, *nlgn3*, and decreased expression of *rab37* and *utrn* detected by transcriptomics was not validated with RT-qPCR; no significant differences were seen between tau-KO and WT mice. *grip2* mRNA expression was altered in the opposite direction.

Previous works have shown the deregulation of some selected mRNAs in tau-KO mice, including the *smarce1* gene, which codes for the BAF-57 protein, a protein involved in the repression of neuron-specific genes, in 3 month-old mice [56], and the gene coding for Gem GTPase, a small GTP-binding protein of the Ras superfamily, in 2 month-old mice [55]. We have not detected the deregulation of these genes in our study. However, our data indicate for the first time a whole image of the transcriptomic status of tau-KO animals with non-apparent phenotypic-significant alterations, demonstrating underlying alterations not previously reported, and suggesting changes in coding RNAs that are functionally linked to neurotransmission systems and synaptic components.

Protein phosphorylation is one of the most common and essential reversible post-translational modifications that modulate protein-protein interactions, protein conformation, and protein signaling, thus regulating protein function [61,62,63,64,65,66]. Our phosphoproteomics study reveals 219 deregulated phosphosites, 121 hypophosphorylated, and 98 hyperphosphorylated. Most deregulated phosphoproteins (*n* = 129) are categorized as cytoskeletal (*n* = 45) and membrane proteins (*n* = 84), including proteins of the synapses and vesicles, myelin proteins, and proteins linked to membrane transport and ion channels. This classification is instrumental, as some membrane proteins also participate in the structure of the synapses, and several synaptic proteins may also be considered plasma membrane proteins.

Other groups of deregulated phosphoproteins are related to DNA and RNA metabolism (*n* = 36), the ubiquitin-proteasome system (UPS) (*n* = 7), and kinases and phosphatases (*n* = 21). Finally, twenty-two deregulated phosphoproteins participate in metabolic pathways, mitochondrial function, and cell growth. Since no commercial antibodies are available for most phosphorylated proteins, we have only validated the deregulated phosphorylation of neurofilaments with Western blotting in tau-KO mice. According to the BioGRID database [67], 324 tau protein interactors have been experimentally demonstrated in mice. Interestingly, 5.5% of this interactome is differentially phosphorylated in tau-KO mice (Stub1, Syngap1, Dbn1, Pacsin1, Sgip1, Srp72, Snap91, Dclk1, Gpm6a, Ppp3ca, Synj1, Mapre2, Pcbp1, Cldn11, Hsph1, Npepps, Ppp3cb, and Tppp). Phosphoproteomics reveals new putative effects of tau ablation that may link deregulated phosphorylation with protein interactions and functions in tau-KO mice.

In addition, a few proteins were assessed with Western blotting to compare mRNA expression with corresponding protein levels. *map1a* mRNA and protein levels were decreased, and increased *tubb3* and *gad1* mRNA were accompanied by reduced β-tubulin III and GAD1 protein levels in tau-KO mice. Differences between RNA and protein expression may be related to modulated translation in tau-KO mice. In another study, proteomics of the frontal cortex revealed altered expression of proteins linked to neuritogenesis in tau-KO mice aged 14 months [42]. Proteomic studies of synaptosomal-enriched fractions of stressed tau-KO mice revealed increased protein levels of 23 mitochondrial proteins compared with stressed non-tau-KO mice [39].

Considered in combination with transcriptomics, phosphoproteomics, and current methods of mRNA and protein validation, the present study provides a comprehensive molecular phenotype of tau-KO mice. Lack of tau protein, as expected, mainly affects by changing the cytoskeleton composition [68,69]. However, other vital structures are also altered in tau-KO mice, including cell membranes, synapses, transmitter receptors, and extracellular matrix. Additional effects are putative functional alterations of DNA and RNA metabolism and cell signaling linked to deregulated phosphorylation. Finally, modified activation or inhibition of kinases and phosphatases may drive the reshaped protein phosphorylation in tau-KO mice.

Each tau isoform is present in the human central nervous system and is believed to have different physiological functions. 3R tau predominates in brain development, but 4R tau increases during early life and reaches levels equal to 3R in adulthood. In the case of rodents, 4R tau increases during development and predominates in the adult brain, whereas 3R tau almost disappears [4,70]. In addition to this differential developmental regulation of alternative splicing in adult human and rodent brains, there are region- and cell-specific patterns of isoform expression, particularly with respect to sequences encoded by exon 10 [11,57,71]. However, the spatial regulation patterns and regulatory mechanisms that control isoform complement in specific cells and nuclei in the brain are not well understood. Thus, different regions might be more susceptible than others to tau’s ablation depending on the region-specific tau expression pattern. Given the alteration of a high number of proteins involved in synapse generation, functioning, and structural maintenance in tau-KO mice, it is likely that their phenotype relies on altered cortical connectivity with other brain structures. For instance, an altered connectivity between the cortex and key nuclei involved in arousal, such as the locus coeruleus, or regulatory centers of risk assessment, such as the nucleus accumbens and the amygdala, may explain why these animals tend to downplay external threats in risk/reward assessment. However, elucidating the functional impact of this set of biochemical alterations will require further research.

Regarding the possible applicability of our results, the reported changes may prevent the development of depressive disorders through alterations in neuronal circuit composition [72,73]. In addition, reducing tau levels has been used as a therapeutic approach in various studies of tauopathies [38,74], and our results have reinforced the relative safety of tau ablation in this type of study. Consistent with this, *mapt* knock-out has also demonstrated its ability to disable seeding and spreading mechanisms in the tau-KO model inoculated with tau-PHFs, an enriched fraction from AD brains [57]. Based on these last approaches, and in parallel to this study, Ionis, in partnership with Biogen Inc., Cambridge, and Massachusetts, has developed the first-ever trial of a tau antisense oligonucleotide (ASO) in mild Alzheimer’s disease. In this Phase 1b trial, BIIB080 caused no serious adverse events. It reduced both total tau and phosphotau-181 in the cerebrospinal fluid by 30–50 percent. However, what this means for cognition remains unclear. A Phase 2 trial is planned to begin in mid-2022 [75]. These experimental clinical trials are encouraging for our study and for further understanding the effects of protein tau lacking in the brain, as well as the consequences of ASOs or siRNA-based therapies. However, more studies need to be conducted to gain more knowledge about this topic. Pursuing this field of study might be very promising in terms of future therapies for neurodegenerative diseases.

## 4. Materials and Methods

### 4.1. Animals

We utilized C57BL/6 mice (wild-type: WT) and homozygous B6.Cg-Mapttm1(EGFP)Klt (mapt knockout: tau-KO) (Jackson Laboratory; Bar Harbor, ME, USA) aged 12 months. tauGFP knock-in/knock-out mice have an EGFP coding sequence inserted into the first exon of the microtubule-associated protein tau (mapt) gene, disrupting expression of mapt and producing a cytoplasmic EGFP fused to the first 31 amino acids (https://www.jax.org/strain/029219, accessed on 27 March 2023). For molecular studies, naïve animals were used. The number of mice was nine per group, with an equal number of males and females. Transgenic mice were identified by genotyping genomic DNA isolated from tail clips using the polymerase chain reaction conditions indicated by the Jackson Laboratory (Bar Harbor). Animals were maintained under standard animal housing conditions in a 12 h dark-light cycle with free access to food and water. All animal procedures were carried out following the guidelines of the European Communities Council Directive 2010/63/EU and with the approval of the local ethical committee (C.E.E.A., Comitè Ètic d’Experimentació Animal; University of Barcelona, Barcelona, Spain; ref. 426/18). Animals were killed by cervical dislocation, and their brains were rapidly removed and processed for study. The left cerebral hemisphere was dissected on ice-chilled plates, immediately frozen, and stored at −80 °C until used for biochemical analyses. The right hemisphere, brainstem, and cerebellum were fixed in 4% paraformaldehyde, cut into coronal sections, and embedded in paraffin. De-waxed sections four microns thick were stained with hematoxylin and eosin or processed for immunohistochemistry.

### 4.2. Mice Behavioral Evaluation Tests

To perform behavioral evaluation tests, we used 8 wild-type animals and 10 tau-KO animals at 12 months of age:-Cognitive performance (two-object recognition test): This paradigm consists of placing the animals for 9 min in a V-maze containing two identical objects at the ends of the arms. Twenty-four hours after the training session, the animals are placed in the V-maze where one of the two familiar objects is replaced by a novel object. The time that the animals spend exploring the two objects is recorded in the training and test sessions; the object Recognition Index (RI) is calculated as the difference between the time spent exploring the novel (TN) and the familiar object (TF) during the test session, divided by the total time spent exploring the two objects [RI = (TN − TF)/(TN + TF)]. Animals exhibiting memory impairments revealed a lower object RI;-Fear conditioning: A contextual fear conditioning procedure was carried out with a computerized StartFear system (Panlab-Harvard, Barcelona, Spain). Shocks were delivered and controlled using Freezing v1.3.04 software (Panlab-Harvard). During fear and conditioning testing, the fear chamber consisted of a black methacrylate box with a transparent front door (25 *×* 25 *×* 25 cm) inside a sound-attenuating chamber (67 *×* 53 *×* 55 cm). During fear conditioning, mice were placed in the fear chambers for 5 min and then received two-foot shocks (1 s, 0.3 mA) separated by a one minute resting period. After the last shock, mice were left in the chamber for another minute. The association between the context and the shock was assessed 24 h after the conditioning during the test session. Mice were placed in the fear chambers for 5 min, and freezing behavior, a rodent’s natural response to fear, was automatically scored. Freezing was defined as the absence of movement except for respiration, and freezing scoring was carried out by a high-sensitivity weight transducer system at the bottom of the experimental chambers that records and analyzes the signal generated by the mouse’s movement. Mice were considered to have frozen when remaining immobile for more than 2000 ms. Episodes were averaged in 60 s slots using Freezing 911 v1.3.04 software (Panlab-Harvard);-Emotional Evaluation: Elevated Plus Maze: The Elevated Plus Maze assesses animals’ anxiety-like behavior. It involves placing the mice in a cross-shaped maze elevated 40 cm above the ground. Two arms of the maze are open laterally so that the animal can perceive the elevation of the labyrinth. Animals presenting lower anxiety levels exhibit higher open-arms exploration scores.

### 4.3. RNA Extraction

RNA purification from the mouse cortex was carried out with the RNeasy Lipid Tissue Mini Kit (Qiagen, Hilden, Germany), following the protocol provided by the manufacturer. The quality of isolated RNA was first measured with the Bioanalyzer Assay (Agilent, Santa Clara, CA, USA). The concentration of each sample was obtained from A260 measurements with Nanodrop 1000 (Thermo Scientific, Wilmington, DE, USA). RNA integrity was tested using the Agilent 2100 BioAnalyzer (Agilent Technologies, Palo Alto, CA, USA).

### 4.4. Transcriptomic Analysis

The transcriptomic analysis of the cerebral cortex of tau-KO animals and controls aged 12 months (*n* = 5 per group) was carried out using RNA-seq technology. RNA libraries were prepared using the NEBNext^®^ Ultra II RNA Library Prep Kit (New England Biolabs) following the manufacturer’s protocol. A total of 25 samples of RNA were validated with an RNA High Sensitivity Tape Station (Agilent Technologies). A quantity of 700 ng of RNA was used to prepare the libraries, which were validated using a DNA 1000 Tape Station Kit. Equimolar proportions of each library were mixed, and the subsequent pool was quantified by qPCR and sequenced in 2 NextSeq High Output 2 *×* 75 run (Illumina). Counts were obtained with HTSeq software (version 2.0.2). Reads were mapped using STAR software against the mouse reference GRCm38 and the GENCODE annotation vM23. Quality control was performed using FASTQC while studying running variables and group cofactors [76,77]. The protein-coding genes were selected for further analysis. At the expression level, no deviation was observed. A filter of lowly expressed genes by the mean values of each gene’s expression (log2CPM > 1) was additionally performed. Genes with low expression were filtered out for further analysis. Finally, we fitted a linear regression model between groups to obtain differentially regulated genes. Then, the *p*-values for the coefficient of interest were adjusted for multiple testing. This method, which controls the expected false discovery rate (FDR) below the specified value, is the default adjustment method because it is the most likely to be appropriate for microarray studies, finally obtaining the significant adjusted *p*-values (*p* < 0.05) for each gene.

### 4.5. Phosphoproteomic Analysis

Brain samples of the left anterior hemisphere from WT and tau-KO mice (*n* = 4 per group) were homogenized in a lysis buffer containing 7 M urea, 2 M thiourea, and 50 mM DTT supplemented with protease and phosphatase inhibitors. The homogenates were spun down at 100,000× *g* for 1 h at 15 °C. Protein quantification was performed with the Bradford assay kit (BioRad, Barcelona, Spain). The phosphoproteomes and the corresponding proteomes were independently analyzed by conventional label-free [78] and SWATH-MS (sequential window acquisition of all theoretical fragment ion spectra mass spectrometry) [79], respectively:-Label-free phosphoproteomics: A quantity of 600 µg of protein was used to obtain phosphorylated fractions. The reduction was performed for protein digestion by adding dithiothreitol (DTT) to a final concentration of 10 mM and incubating at RT for 30 min. Subsequent alkylation with 30 mM (final concentration) iodoacetamide was performed for 30 min in the dark at room temperature. An additional reduction step was performed with 30 mM DTT (final concentration), allowing the reaction to stand at room temperature for 30 min. The mixture was diluted to 0.6 M urea using MilliQ water, and after the addition of trypsin (Promega, Madison, WI, USA) (enzyme:protein, 1:50 w/w), the sample was incubated at 37 °C for 16 h. Digestion was quenched by acidification (pH < 6) with acetic acid. After protein enzymatic cleavage, peptide cleaning was performed using Pierce™ Peptide Desalting Spin Columns (ThermoFisher Scientific, Waltham MA, USA). The High-Select™ TiO2 Phosphopeptide Enrichment Kit (ThermoFisher Scientific, Waltham, MA, USA) was used to obtain the phosphorylated peptide fractions according to the manufacturer’s instructions. Phosphopeptide mixtures were separated by reversed-phase chromatography using an Eksigent NanoLC ultra-2D pump fitted with an Acclaim™ PepMap™ 100 C18 column (0.075 × 250 mm, particle size 3 µm; ThermoFisher Scientific, Waltham, MA, USA). Samples were first loaded for concentration into an Acclaim™ PepMap™ 100 C18 trap column (0.1 × 20 mm, particle size 5 µm; ThermoFisher Scientific, Waltham, MA, USA). Mobile phases were 100% water, 0.1% formic acid (FA) (buffer A), and 100% acetonitrile with 0.1% FA (buffer B). A column gradient was developed from 2% B to 40% B in 120 min. The column was equilibrated at 95% B for 10 min and 2% B for 10 min. The pre-column was aligned with the column, and flow was maintained along the gradient at 300 nL/minute. Eluting peptides were analyzed using a 5600 Triple-TOF mass spectrometer (Sciex, Framingham, MA, USA). Informational data acquisition was obtained upon a survey scan performed in a mass range from 350 m/z up to 1250 m/z in a scan time of 25 ms. The top 15 peaks were selected for fragmentation. The minimum accumulation time for MS/MS was set to 200 ms, giving a total cycle time of 3.3 s. Product ions were scanned in a mass range from 100 m/z up to 1500 m/z and excluded for further fragmentation for 15 s. The raw MS/MS spectra searches were processed using the MaxQuant software (v 1.6.7.0) [80] and searched against the Uniprot proteome reference for Mus Musculus (Proteome ID: UP000000_10090, March 2021). The parameters used were as follows: initial maximum precursor (15 ppm); fragment mass deviations (20 ppm); fixed modification (Carbamidomethyl (C)); variable modification (Oxidation (M)); acetyl (protein N-terminal; phospho (STY)); enzyme (trypsin) with a maximum of 2 missed cleavages; minimum peptide length (7 amino acids); and false discovery rate (FDR) for PSM and protein identification (1%). Frequently observed laboratory contaminants were removed. The Perseus software (version 1.6.14.0) [81] was used for statistical analysis and data visualization;-SWATH-MS: For protein expression analysis, a pool containing the same amount of µg per sample was used as input for the generation of the SWATH-MS assay library. Twenty micrograms were diluted in Laemmli buffer and loaded into a 0.75 mm-thick polyacrylamide gel with a 4% stacking gel cast over a 12.5% resolving gel. The total gel was stained with Coomassie Brilliant Blue, and 12 equal slides from each pooled sample were excised from the gel and transferred into 1.5 mL Eppendorf LoBind tubes. Protein enzymatic cleavage was carried out with trypsin (1:20, *w*/*w*, Promega, Madison, WI, USA) at 37 °C for 16 h, as previously described [82]. Purification and concentration of peptides were performed using C18 Zip Tip Solid Phase Extraction (Millipore, Burlington, MA, USA). The peptides, recovered from in-gel and in-solution digestion processing, were reconstituted into a final concentration of 0.5 µg/µL of 2% ACN, 0.5% FA, and 97.5% MilliQ water before mass spectrometric analysis. MS/MS datasets for spectral library generation were acquired on a Triple TOF 5600+ mass spectrometer (Sciex) interfaced to the Eksigent nanoLC ultra-2D pump system (Sciex) as previously described. MS/MS data acquisition was performed using AnalystTF 1.7 (Sciex), and spectra files were processed through the ProteinPilot v5.0 search engine (Sciex) using the ParagonTM Algorithm (v.4.0.0.0) [83] for database search. The identified proteins were grouped based on MS/MS spectra by the Progroup™ Algorithm, regardless of the peptide sequence assigned. The false discovery rate (FDR) was determined using a non-linear fitting method [84], and the displayed results were those reporting a 1% Global FDR or better. Then, individual protein extracts from all sample sets (*n* = 10) were subjected to in-solution digestion, peptide purification, and reconstitution before mass spectrometric analysis. Protein extracts (20 g) from each sample were reduced by adding DTT to a final concentration of 10 mM and incubation at room temperature. The reduction was performed by adding DTT to a final concentration of 10 mM and incubating at room temperature for 30 min. Subsequent alkylation by 30 mM iodoacetamide was performed for 30 min in the dark. An additional reduction step was performed with 30 mM DTT, allowing the reaction to stand at room temperature for another 30 min. The mixture was then diluted to 0.6 M urea using MilliQ water, and after trypsin addition (Promega, Madison, WI, USA) (enzyme:protein, 1:50 *w*/*w*), the sample was incubated at 37 °C for 16 h. Digestion was quenched by acidification with acetic acid. The digestion mixture was dried in a SpeedVac. Purification and concentration of peptides were performed using C18 Zip Tip Solid Phase Extraction (Millipore, Burlington, MA, USA). The peptides recovered were reconstituted into a final concentration of 1 µg/µL of 2% ACN, 0.5% FA, and 97.5% MilliQ water before mass spectrometric analysis. For SWATH-MS-based experiments, the TripleTOF 5600+ instrument was configured as described in [85]. Using an isolation width of 16 Da (15 Da for optimal ion transmission efficiency and 1 Da for the window overlap), a set of 37 overlapping windows was constructed, covering the mass range 450–1000 Da. Using this method, 1 μL of each sample was loaded onto an Acclaim™ PepMap™ 100 C18 trap column (0.1 × 20 mm, particle size 5 µm; ThermoFisher Scientific, Waltham, MA, USA) and desalted with 100% water and 0.1% formic acid at 2 μL/minute for 10 min. The peptides were loaded onto an Acclaim™ PepMap™ 100 C18 column (0.075 × 250 mm, particle size 3 µm; ThermoFisher Scientific, Waltham, MA, USA) equilibrated in 2% acetonitrile and 0.1% FA. Peptide elution was carried out with a linear gradient of 2%–40% B for 120 min (mobile phases A: 100% water, 0.1% formic acid (FA), and B: 100% acetonitrile, 0.1% FA) at a flow rate of 300 nL/minute. Eluted peptides were infused into the mass spectrometer. The Triple TOF was operated in SWATH mode, in which a 0.050 s TOF MS scan from 350 to 1250 m/z was performed, followed by 0.080 s product ion scans from 230 to 1800 m/z on the 37 defined windows (3.05 s/cycle). The collision energy was set to optimum for a 2+ ion at the center of each SWATH block with a 15 eV collision energy spread. The resulting ProteinPilot group file from library generation was loaded into PeakView^®^ (v2.1, Sciex), and peaks from SWATH runs were extracted with a peptide threshold of 99% confidence (Unused Score ≥ 1.3) and a FDR lower than 1%. For this, the MS/MS spectra of the assigned peptides were extracted by ProteinPilot, and only the proteins that fulfilled the following criteria were validated: (1) peptide mass tolerance lower than 10 ppm, (2) 99% confidence level in peptide identification, and (3) complete b/y ion series found in the MS/MS spectrum. Only proteins quantified with at least two unique peptides were considered. Then, quantitative data were analyzed using the Perseus software (version 1.6.14.0) for statistical analysis and data visualization. The two-sample Student’s t-test was applied to compare between groups. Then, only phosphopeptides with a *p*-value < 0.05 were considered differentially expressed. Proteomic experiments generate a large number of peptides or proteins that need to be independently evaluated using statistical tests and may yield type I errors [86]. However, it is important to note that, due to the often low power of proteomic experiments, the use of these corrections may fail to detect even true positives [87]. In this case, the use of only five samples per group, together with the low fold changes quantified in our data, were determinants for the statistical analysis, and as a consequence, using FDR corrections was not able to detect any significant phosphopeptide. MS data and search result files were deposited in the Proteome Xchange Consortium via the JPOST partner repository (https://repository.jpostdb.org, accessed on 1 March 2022) [88] with the identifiers PXD040416 for ProteomeXchange and JPST002064 for jPOST. (https://repository.jpostdb.org/preview/165924140563fcaaf142e99; Access key 9943);-Bioinformatics: The identification of significantly deregulated regulatory/metabolic pathways in proteomic datasets was performed using ShinyGO 0.77 [89].

### 4.6. Retrotranscription Reaction-qPCR

The retrotranscriptase reaction was carried out using a High-Capacity cDNA Archive Kit (Applied Biosystems, Foster City, CA, USA) following the protocol provided by the supplier. Parallel reactions for an RNA sample were run without MultiScribe reverse transcriptase to assess the degree of contaminating genomic DNA. TaqMan quantitative RT-PCR assays for each gene were performed in duplicate on cDNA samples in 384-well optical plates using an ABI Prism 7900 Sequence Detection System (Applied Biosystems, Life Technologies, Waltham, MA, USA). Nine samples from wild-type animals and eight samples from tau-KO animals were used for RT-qPCR. For each 10 μL TaqMan reaction, 4.5 μL of cDNA was mixed with 0.5 μL 20× TaqMan Gene Expression Assays and 5 μL of 2× TaqMan Universal PCR Master Mix (Applied Biosystems). We used the internal housekeeping genes, alanyl-tRNA synthase (*aars*), glucuronidase B (*gusB*), hypoxanthine phosphoribosyltransferase 1 (*hprt1*), and X-prolyl aminopeptidase (aminopeptidase P) 1 (*xpnpep1*) for normalization. The reactions were carried out using the following parameters: 50 °C for 2 min, 95 °C for 10 min, and 40 cycles of 95 °C for 15 s and 60 °C for 1 min. Finally, all TaqMan PCR data were captured using the Sequence Detection Software (SDS version 1.9; Applied Biosystems). The identification numbers and names of all TaqMan probes used are shown in Table 1. The selection of these particular probes was based on RNA-seq results. Samples were analyzed with the double-delta cycle threshold (ΔΔCT) method. The ΔCT values represent normalized target gene levels for the internal control. The ΔΔCT values were calculated as the ΔCT of each test sample minus the mean ΔCT of the calibrator samples for each target gene. The fold change was determined using equation 2(−ΔΔCT).

### 4.7. Gel Electrophoresis and Western Blotting

Frozen samples of the posterior part of the left hemisphere from 7 WT and seven tau-KO mice were homogenized in RIPA lysis buffer composed of 50 mM Tris/HCl buffer, pH 7.4, containing 2 mM EDTA, 0.2% Nonidet P-40, 1 mM PMSF, and a protease and phosphatase inhibitor cocktail (Roche Molecular Systems, Pleasanton, CA, USA). The homogenates were centrifuged for 20 min at 12,000 rpm. Protein concentration was determined with the bicinchoninic acid (BCA) method (Thermo Scientific, MA, USA). Equal amounts of protein (12 μg) for each sample were loaded and separated by electrophoresis on 10% sodium dodecyl sulfate polyacrylamide gel electrophoresis (SDS-PAGE) gels and transferred onto nitrocellulose membranes (Amersham, Freiburg, Germany). Non-specific bindings were blocked by incubation with 3% albumin in phosphate-buffered saline (PBS) containing 0.2% Tween for 1 h at room temperature. After washing, membranes were incubated overnight at 4 °C with antibodies against different forms of tau protein (Table 2). Protein loading was monitored using an antibody against β-actin (42 kDa, 1:30,000, Sigma-Aldrich, San Luis, MO, USA) or vinculin (125 kDa, sc-73614, 1:1000, Santa Cruz Biotech, Dallas, TX, USA) when β-actin molecular weight was overlapped or out of range. Membranes were incubated for 1 h with appropriate HRP-conjugated secondary antibodies (1:3000, Dako, Glostrup, Denmark); the immunoreaction was revealed with a chemiluminescence reagent (Amersham, Freiburg, Germany). Results were analyzed statistically with SPSS 19.0 (SPSS Inc., NY, USA) and GraphPad Prism (GraphPad Software, Inc. version 5.01) software. Data were presented as mean ± standard error of the mean (SEM). The unpaired Student’s t-test was used to compare each group. The significance level was set at * *p* < 0.05, ** *p* < 0.01, and *** *p* < 0.001.

### 4.8. Statistics

Differences between wild-type and tau-KO animals in q-PCR, Western blot, and behavioral data were analyzed with the Student’s t-test using the SPSS software (IBM Corp., released 2013, IBM-SPSS Statistics for Windows, Version 19.0, Armonk, NY, USA). Graphic design was performed with GraphPad Prism version 5.01 (La Jolla, CA, USA). Outliers were detected using the GraphPad software QuickCalcs (version 9.0) (*p*  <  0.05) (La Jolla, CA, USA). The data were expressed as mean ± SEM, and significance difference levels between tau-KO and WT mice were set at * *p* < 0.05, ** *p* < 0.01, and *** *p* < 0.001.

## Figures and Tables

**Figure 1 ijms-24-06559-f001:**
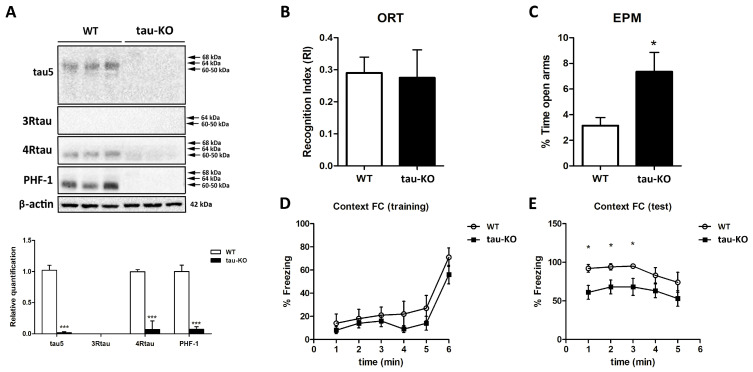
(**A**) Western blots of brain homogenates of WT mice show immunoreactive bands at the appropriate molecular weight for antibodies tau 5, 4R tau, and PHF1; Western blots for 3R tau are negative. As expected, Western blots to tau 5, 4R tau, and PHF1 (and 3R tau) of brain homogenates from tau-KO mice are negative. Molecular weights pointed with arrows indicate the theoretical positions of all tau isoform bands in an immunoblotting membrane. A densitometric study of the bands confirms significant differences between WT and tau-KO mice, *** *p* < 0.001. (**B**) A two-object recognition test reveals no differences between WT and tau-KO mice. (**C**) The results of the elevated plus maze reveal a reduced anxiety-like behavior in tau-KO mice, spending more time in the open arms compared to WT (*p* < 0.05). (**D**) No differences are seen in the training trial of the contextual fear conditioning test. (**E**) However, transient but significantly less freezing occurs in the first 3 min of re-exposure to the aversively conditioned context in tau-KO mice (* *p* < 0.05).

**Figure 2 ijms-24-06559-f002:**
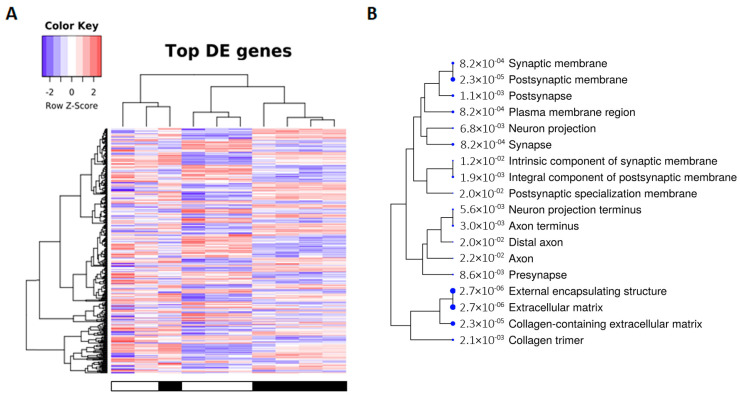
(**A**) Transcriptomic analysis shows differentially expressed (DE) genes in tau-KO mice compared with WT. After filtering, 69 were downregulated (purple tones), and 105 were upregulated (red tones). Deregulated transcripts are graphically illustrated in the heat map representation at a probability of an adjusted *p*-value < 0.05. Sample legend in the heat-map plot: white boxes correspond to wild-type animals, and black boxes correspond to tau-KO animals. (**B**) Enrichment analysis against the Gene Ontology database (*p* < 0.05) identifies primary altered genes corresponding to (i) synaptic structures, (ii) neuron cytoskeleton and transport, and (iii) extracellular matrix components. The blue dot size beneath each gene cellular component category is proportional to the enrichment FDR score number (between dot and category name) obtained from the differentially expressed genes involved in the functional group.

**Figure 3 ijms-24-06559-f003:**
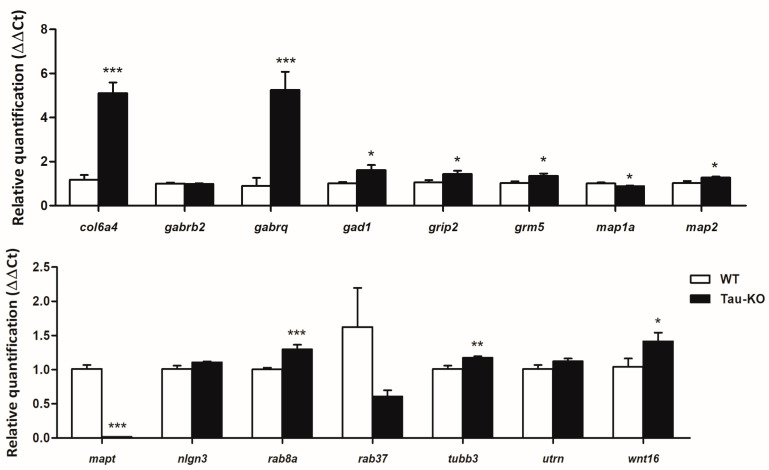
RT-qPCR of the cerebral cortex of tau-KO and WT mice aged twelve months. Tau-KO mice show increased mRNA levels of *col6a4*, *gabrq*, *gad*, *grm5*, *grip2*, *map1a*, *map2*, *rab8a*, *tubb3*, and *wnt16* compared with WT mice. However, the expression of *gabrb2*, *nlgn3*, *rab37*, and *utrn* is similar in tau-KO and WT mice. As expected, *mapt* mRNA expression is 0 in tau-KO mice. * *p* < 0.05, ** *p* < 0.01, and *** *p* < 0.001.

**Figure 4 ijms-24-06559-f004:**
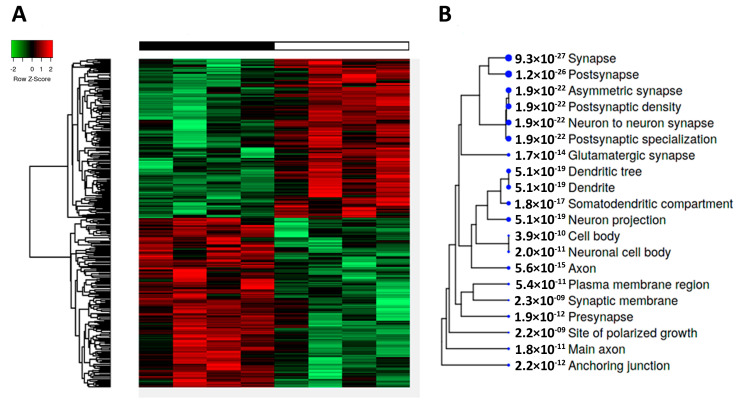
(**A**) A heatmap of the cerebral cortex’s phosphoproteome shows 121 hypophosphorylated and 98 hyperphosphorylated proteins. (**B**) The altered phosphoproteome in tau-KO mice mainly affects synapses, axons, dendrites, and the neuronal cell body. Sample legend in the heat-map plot: white boxes correspond to wild-type animals, and black boxes correspond to tau-KO animals. The blue dot size beneath each gene cellular component category is proportional to the enrichment FDR score number (between dot and category name) obtained from the differentially expressed proteins involved in the functional group.

**Figure 5 ijms-24-06559-f005:**
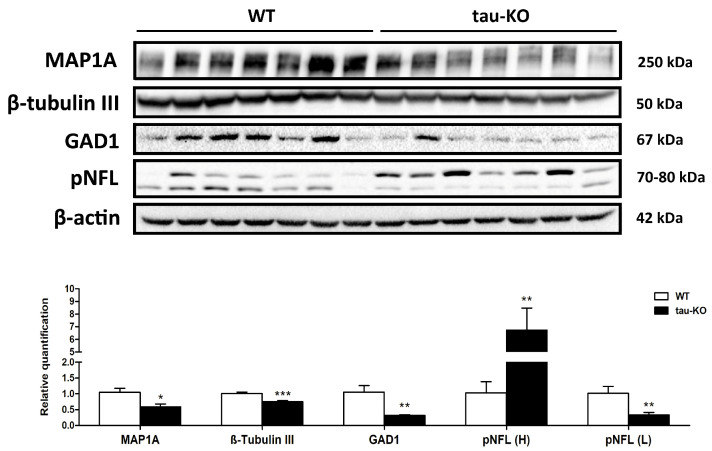
Western blotting of the cerebral cortex of tau-KO and WT mice aged twelve months. A significant reduction of MAP1A, β-tubulin III, and GAD1 is found in tau-KO mice. Phosphorylated neurofilament heavy chain (pNFL-H) is increased, and phosphorylated neurofilament light chain (pNFL-L) is decreased in tau-KO mice. * *p* < 0.05, ** *p* < 0.01, and *** *p* < 0.001.

**Table 1 ijms-24-06559-t001:** List of TaqMan probes used to assess transcript levels by RT-qPCR.

Gene	Gene Symbol	Reference
Collagen alpha-4(VI) chain	*col6a4*	Mm01231949_m1
Gamma-aminobutyric acid type A receptor subunit beta2	*gabrb2*	Mm00433467_m1
Gamma-aminobutyric acid type A receptor subunit tyheta	*gabrq*	Mm00445057_m1
Glutamate decarboxylase 1	*gad1*	Mm00725661_s1
Glutamate metabotropic receptor 5	*grm5*	Mm00690332_m1
Glutamate receptor-interacting protein 2	*grip2*	Mm01183453_m1
Microtubule associated protein 1A	*map1a*	Mm01330378_m1
Microtubule associated protein 2	*map2*	Mm00485231_m1
Microtubule associated protein tau	*mapt*	Mm00521988_m1
neuroligin-3	*nlgn3*	Mm01225951_m1
RAB37, member RAS oncogene family	*rab37*	Mm00445351_m1
Ras-related protein Rab-8A	*rab8a*	Mm00445684_m1
Tubulin beta 3 class III	*tubb3*	Mm00727586_s1
Utrophin	*utrn*	Mm01168866_m1
Wnt family member 16	*wnt16*	Mm00446420_m1
alanyl-tRNA synthetase	*aars*	Mm00507627_m1
Hypoxanthine-guanine phosphoribosyltransferase	*hprt-1*	Mm01545399_m1
X-prolyl aminopepidase P1	*xpnpep1*	Mm00460040_m1
β-glucuronidase	*gusβ*	Mm01197698_m1

**Table 2 ijms-24-06559-t002:** List of antibodies used to assess protein levels by Western blotting, including the antibody’s commercial reference, the antibody’s production host, and the used Western blot dilution.

Antibody	Supplier	Reference	Host	Dilution WB
Tau 5	ThermoFisher	MA5-12808	Ms	1/500
3R tau	Millipore	05-803	Ms	1/1000
4R tau	Millipore	05-804	Ms	1/1000
PHF-1	Dr. Peter Davies	-	Ms	1/1000
β-actin	Sigma	A5316	Ms	1/30,000
map1a	Millipore	MAB362	Ms	1/500
tubulin b-III	Signalway	21617	Rb	1/1000
pNFL (Ser473)	Millipore	MABN2431	Ms	1/500
GAD1	Cell Signaling	#5305	Rb	1/200
vinculin	Santa Cruz	Sc-73614	Ms	1/1000

## Data Availability

All the supporting data and access information are included in the manuscript.

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
