# Peer review of "Deregulated Transcription and Proteostasis in Adult mapt Knockout Mouse"

_ijms, 2023, doi:10.3390/ijms24076559_

Round 1

Reviewer 1 Report

The authors performed transcriptomics and phosphoproteomic analysis using B6.Cg-Mapttm1(EGFP)Klt (tau-KO) and wild -type mice. Their tau-KO mice showed subtle behavioral change in fear conditioning and two object recognition test suggesting anxiety-like behavior. Trascriptomic analysis by RNAseq revealed change in synaptic structure, neuron cytoskeleton and extracellular matrix components. Phosphoproteomic analysis showed change in synapses, axons, dendrites and neuronal cell body. They confirmed some of those using qPCR or western blotting.

The result is exploratory, thus acceptable in any context.

Major concern.

1)    In RNAseq analysis or in phosphoproteomic analyses, did the authors perform multiple comparison test? Data presentation doesn’t look like they did, thus their discovery might be false discovery.

Minor concern

1)    As the authors cite, adult mice tau is 4R whereas human tau being 3R+4R. The authors are advised to discuss this issue when translating behavioral change.

Author Response

Dear reviewer, thanks for your comments, we think that have improved the manuscript quality. Find in the attached file the response to your comments with the changes carried out. Changes are indicated in green in the draft.

Comments and Suggestions for Authors: The authors performed transcriptomics and phosphoproteomic analysis using B6.Cg-Mapttm1(EGFP)Klt (tau-KO) and wild -type mice. Their tau-KO mice showed subtle behavioral change in fear conditioning and two object recognition test suggesting anxiety-like behavior. Trascriptomic analysis by RNAseq revealed change in synaptic structure, neuron cytoskeleton and extracellular matrix components. Phosphoproteomic analysis showed change in synapses, axons, dendrites and neuronal cell body. They confirmed some of those using qPCR or western blotting. The result is exploratory, thus acceptable in any context.

Major concern

1)    In RNAseq analysis or in phosphoproteomic analyses, did the authors perform multiple comparison test? Data presentation doesn’t look like they did, thus their discovery might be false discovery.

  • In RNA-seq analysis, multiple comparison test was performed in the initial version of this manuscript (adjusted p-value), but the explanation was not included in materials and methods section. Once revised, the explanation has been included in materials and methods section.
  • In the phosphoproteomics experiment, statistical analysis was performed using the Perseus software. Since the study was only performed comparing two different groups (WT and Tau KO), the two sample Student T test was the suitable choice in this case. Then, only phosphopeptides with a p value < 0,05 were considered differentially expressed. In this sense, we are completely aware that proteomic experiments generate a large number of peptide or proteins that need to be evaluated independently using statistical tests maybe yielding type I errors. Thus, FDR calculations might provide a more accurate measure preventing the appearance of false positives (Nakayasu E et al. Nature protocols, 2021; https://pubmed.ncbi.nlm.nih.gov/34244696/). However, it is important to note that due to the often-low power of proteomic experiments, the use of these corrections may fail to detect even true positives (Pascovici D et al. Proteomics 2016; https://pubmed.ncbi.nlm.nih.gov/27461997/). In this case, the use of only five samples per group together with the low fold changes quantified in our data, were determinant for the statistical analysis and using FDR corrections, we were not able to detect any significant phosphopeptide. This justification and explanation have been included in materials and methods section.

Minor concern

1)    As the authors cite, adult mice tau is 4R whereas human tau being 3R+4R. The authors are advised to discuss this issue when translating behavioral change. -> Lack of mapt gene implies the ablation of any of these isoforms in mouse brain, so we cannot differentiate between isoforms expressions and presence. However, there are some differences between regions, cell and neuronal patterns. Based on that, we discussed about the relationship between the deregulated transcription and proteostasis and the reduced anxiety-like behavior and lower fear expression in the discussion section.

Reviewer 2 Report

In this study, the authors report a comprehensive molecular phenotype of tau-KO mice using techniques combining transcriptomics, (phospho)proteomics, and current methods of mRNA and protein validation. There observations reveal a complex altered brain transcriptome and phosphoproteome in tau-KO mice with only mild behavioral alterations. The manuscript is well-written and organized. The conclusions are scientifically solid and supported by the data. This manu is almost in a good form ready for publication.

Comments:

1.      My suggestion is that the authors should discuss more about the relationship between the deregulated transcription and proteostasis and the reduced anxiety-like behavior and lower fear expression.

2.      What insights can we get on tauopathies from this study of tau-KO mice?

3.      Figure1A, please explain what “68 kD”, “64kD” and “60-50 kD” indicate.  

Author Response

Dear reviewer, thanks for your comments, we think that have improved the manuscript quality. Find in the attached file the response to your comments with the changes carried out. Changes are indicated in green in the draft.

Comments and Suggestions for Authors: In this study, the authors report a comprehensive molecular phenotype of tau-KO mice using techniques combining transcriptomics, (phospho)proteomics, and current methods of mRNA and protein validation. There observations reveal a complex altered brain transcriptome and phosphoproteome in tau-KO mice with only mild behavioral alterations. The manuscript is well-written and organized. The conclusions are scientifically solid and supported by the data. This manu is almost in a good form ready for publication.

Comments:

1. My suggestion is that the authors should discuss more about the relationship between the deregulated transcription and proteostasis and the reduced anxiety-like behavior and lower fear expression. -> We discussed about the relationship between the deregulated transcription and proteostasis and the reduced anxiety-like behavior and lower fear expression in the discussion section.

 2. What insights can we get on tauopathies from this study of tau-KO mice? -> This topic has been added and discussed at the end of the discussion section.

 3. Figure1A, please explain what “68 kD”, “64kD” and “60-50 kD” indicate.  -> It has been indicated in the figure footnote.
